# Composition-Related Dielectric, Ferroelectric and Electrocaloric Properties of Pb_5_Ge_3_O_11_ Single Crystals Modified by Ba Ions

**DOI:** 10.3390/ma16010413

**Published:** 2023-01-01

**Authors:** Magdalena Krupska-Klimczak, Irena Jankowska-Sumara, Przemysław Gwizd, Marceli Koralewski, Andrzej Soszyński

**Affiliations:** 1Department of Exact and Natural Sciences, Pedagogical University of Krakow, Podchorążych 2, 30-084 Kraków, Poland; 2Institute of Security Studies and Computer Science, Pedagogical University of Krakow, Podchorążych 2, 30-084 Kraków, Poland; 3Faculty of Physics, Adam Mickiewicz University in Poznań, Uniwersytetu Poznańskiego 2, 61-614 Poznań, Poland; 4Institute of Physics, University of Silesia, 75 Pułku Piechoty, 41-500 Chorzow, Poland

**Keywords:** ferroelectrics, phase transition, electrocaloric effect

## Abstract

In this paper, we studied some ferroelectric properties of archetypal oxide uniaxial ferroelectric single crystals of Pb_5_Ge_3_O_11_ modified by Ba ions. They include dielectric, DSC, ferroelectric polarization, and electrocaloric effect (ECE) measurements. The measurements show that increasing Ba doping considerably influences all the measured parameters, mainly by lowering the Curie temperature, gradually diffusing the phase transition, and decreasing values of polarization as well as the coercive field. The decrease in overall ECE is influenced by decreasing polarization. Compared with the pure PGO single crystals, this decrease is from 1.2 K to 0.2 K. However, the effect of diffusing the phase transition increases the range of its occurrence (up to 30 K), which might be beneficial in applications.

## 1. Introduction

Lead germanate single crystal Pb_5_Ge_3_O_11_ (PGO) is a well-known oxide uniaxial ferroelectric [1]. A special property of PGO is high optical activity below its Curie temperature (*T*_C_). This ferroelectric material owes its popularity to its excellent electro-optic properties related to its pyroelectric and photorefractive effects [1,2,3]. At *T*_C_ = 450 K, lead germanate undergoes a second-order phase transition from the paraelectric to the ferroelectric phase (*C*_3h_–*C*_3_) [4,5]. Due to the uniaxial structure, the properties of PGO, such as electric permittivity, piezoelectric, electro-optic, and elastic coefficients, are strongly anisotropic. In the literature, one can find many studies on PGO doped with various elements such as Ba, Cr, Sr, Cu, Mn, La, Nb, etc. [1]. Most of them describe mainly electro-optical behavior due to doping. However, several studies dealing with the structure of these materials and their dielectric, piezo- and pyroelectric, mechanical, and other properties can also be found in the literature [6,7,8,9,10]. In contrast, there are few studies on the electro-thermodynamic properties of these materials, such as the electrocaloric effect (ECE).

The ECE is currently at the forefront of interest due to the possibility of constructing miniature cooling elements, especially those used in microelectronics. A special role in electrocaloric devices is played by ferroelectric materials due to the easily switchable spontaneous polarization they possess. Many papers have recently been devoted to studying the electrocaloric effect specifically in ferroelectric materials, e.g., [11,12,13].

The ECE, i.e., the conversion of electrical energy to heat and vice versa, is of great importance for its application in the new generation of cooling and heating devices, which would be more friendly to the environment. The ECE is already a well-known phenomenon, but it is still not useful for application due to its small magnitude in currently known materials. For this reason, scientists are still looking for a material (single crystal, ceramic, or thin film) that would have a relatively large ECE (of several or several dozen degrees).

In our previous paper, we investigated the influence of heterovalent doping by Cr^3+^ ions on the electrocaloric properties of ferroelectric PGO crystals [10]. It was found that Cr ions enhance the electrocaloric properties of PGO crystals via the reduction of electrical conductivity due to Cr^3+^ doping. Encouraged by this fact, we decided to study this effect in PGO doped at this time with isovalent ions, namely Ba^2+^. According to the publication [8], XPS measurements revealed that Ba ions replace Pb ions in the crystal net, whereas Cr^3+^ ions take the place of Ge^4+^ ions [14]. As reported in [8], the addition of barium ions to PGO considerably influences the phase transition temperatures in those crystals. Specifically, as the concentration of Ba^2+^ ions increases, the transition temperature decreases almost linearly. At the same time, the phase transition diffuses and the maximum dielectric permittivity at *T*_C_ lowers. [7,8]. It was also reported that the A-site substitution of the smaller Pb^2+^ cations (1.19 Å [15]) by significantly larger Ba^2+^ ions (1.35 Å [15]) may result in some inhomogeneous local stresses. Such stresses may be a source of polar nanoregions with an effective dipole moment, as was postulated in the papers [8,9]. Simultaneously, it improves the pyroelectric effect, which is beneficial for the applications [16,17]. From this perspective, studying the electrocaloric properties of PGO after doping with isovalent ions of varying sizes is an interesting experience for us.

## 2. Materials and Methods

The single crystals of Ba-doped lead germanate Pb_5 − x_Ba_x_Ge_3_O_11_ were grown using the Czochralski technique at the Royal Signals and Radar Establishment laboratory (Malvern, UK). The details were published elsewhere [16,18]. The amount of Ba admixture was as follows: x = 0.1, 0.2, 0.25, and 0.3 (hereafter in this publication denoted as PGO + xBa or PGO + 0.1Ba, PGO + 0.2Ba, PGO + 0.25Ba, and PGO + 0.3Ba, respectively). The real composition and structural properties of the crystals were determined by the authors of ref [8] based on the XPS and XRD measurements. The as-grown crystals were transparent and light yellow (except for PGO + 0.2 Ba, which had a slightly darker yellow color—see Figure 1). For the specific heat measurements, differential scanning calorimetry (DSC) was used, and the specimens consisted of single pieces of crystal with an average mass of 20 mg and were placed in an alumina crucible. The measurements were conducted using a Netzsch DSC F3 Maia (Netzsch, Selb, Germany) calorimeter in the temperature range of 150–550 K. For electrical measurements, the as-grown crystals were first cut perpendicularly to the c-axis into slices of ~0.5 mm thickness. The orientation of the c-axis was verified using the standard Laue technique [19]. In the next step, the slices were placed on a precision wire saw with an adjustable stage and cut into plates of average dimensions of 3 × 1 × 0.5 mm^3^. Crystal plates oriented along the ferroelectric [001]-axis were obtained in this manner for dielectric property and electric polarization measurements. For electrical measurements, the crystals were covered with silver electrodes and placed in a computer-controlled furnace in which the temperature was controlled with an accuracy of 0.1 K. The dielectric properties were measured in the frequency range from 20 Hz to 2 MHz with an E4980A precision LCR meter (Keysight, Santa Rosa, United States). The electric polarization hysteresis *P*(*E*) measurements were performed using the standard Sawyer–Tower method [20] in a quasistatic limit. The frequency of the sine signal was set to 6 Hz, and the amplitude of the electric field was changed from 0 to 2 × 10^6^ V/m (depending on the exact sample thickness). The measurements were conducted in the heating and cooling modes in the temperature range of 300–450 K. The ECE was calculated by using the Maxwell relations between entropy (*S*) and electric polarization (*P*) [10]. To obtain the reversible changes of electrocaloric temperature change Δ*T* and entropy change Δ*S* as a response to an electric field change, the following equations were used [21]:(1)$$∆T=-\frac{1}{C_{p}\rho }\int_{E_{1}}^{E_{2}}T{\left(\frac{\partial P}{\partial T}\right)}_{E}dE,$$
(2)$$∆S=-\frac{1}{\rho }\int_{E_{1}}^{E_{2}}{\left(\frac{\partial P}{\partial T}\right)}_{E}dE$$
where *P* is polarization, *T* is the temperature, *E*_1_ and *E*_2_ are the initial and final electric fields, and *c_p_* is the value of specific heat for a given temperature.

## 3. Results and Discussion

### 3.1. Dielectric Measurements

The examples of *ε*(*T*) dependences at the measuring frequency *f* = 20 kHz for differently doped samples are presented in Figure 2a. The data on cooling are also presented. The first aspect that must be mentioned when viewing the *ε*(*T*) dependence for PGO samples doped with Ba is the noticeable decrease in the phase transition temperature compared with the pure PGO. The addition of barium ions into PGO also causes a gradual (as Ba content increases) diffusion of the phase transformation as well as a decreasing value of *ε_max_* at the *T*_C_ point. The visible exception is PGO + 0.2Ba, which shows inferior results most likely related to the greater defectiveness of the crystal, i.e., darker coloration. In [16], it has been described that the difference in coloration of the crystals appears when the crystals are grown in the presence of oxygen-contaminated conditions. In the presence of oxygen, the color of the crystal turns out to be darker than in a situation of oxygen deficit. According to XPS measurements presented in [8], PGO crystals always show some degree of oxygen deficit. This deficit varies depending on technological conditions, particularly those described above. In the case of our crystals, all compositions are light yellow, whereas PGO + 0.2Ba is slightly darker yellow. According to [16], the coloration should not exceed the ferroelectric properties of the crystals, but it can influence the conductivity. Higher (or lower) conductivity influences, as a consequence, electric permittivity (*ε*) values. Figure 2b shows the reciprocal 1/*ε* (*T*) to verify the fulfillment of its paraelectric part with the Curie–Weis law. The values of *T*c, ε_max_, temperature *T*_O_, and the Curie–Weiss (*C*) constant resulting from the Curie–Weiss law are presented in Table 1. The values of *T*o are close to *T*c, which confirms the second-order phase transformation in all compositions. The values of the *C* constant are of order 10^4^, which points to the mostly displacive mechanism of the phase transition. It is easy to notice that *T*c decreases almost linearly with the Ba content, which is in agreement with the literature reports [8,17].

Figure 3a presents the temperature dependences of the real part of the linear dielectric susceptibility on heating and cooling modes for one chosen composition (PGO + 0.1Ba). No hysteresis in the thermal cycle was observed. This again confirms the second order of the phase transition. For this sample, we have also observed relatively large dielectric dispersion, which can be seen in Figure 3b. However, this dispersion is not of a relaxation nature—there is no temperature shift of the dielectric permittivity maximum towards higher temperatures as the measurement frequency increases. It has been reported [17] that at low frequencies, electronic, ionic, and interfacial/surface polarization contribute to the electric permittivity. However, for frequencies above 20 kHz, the contributions from the latter can be minimized. At low frequencies, the mobile charges diffuse under the influence of the applied field up to the interface and build up the surface charge until the applied field reverses with the alternating voltage. Intriguingly, for higher-doped lead germanate single crystals, the strength of this dispersion considerably decreases. This indicates that the Ba addition compensates at least partially for the ionic contribution to this dispersion.

Table 1 shows the collected dielectric parameters for PGO single crystals doped with Ba ions in comparison with the data for a pure PGO crystal, which were taken from the paper [10]. The effect of doping on the maximum value of electrical permittivity *ε* can be clearly seen, i.e., a severe decline in the *ε* value for doped crystals in comparison with the pure crystal.

### 3.2. Specific Heat Measurements

In Figure 4, the measurements of specific heat in a wide temperature range of 150–550 K are presented for all considered crystals. The anomalies associated with the phase transitions for second-order phase transitions are usually small and extended, forming a lambda shape. The anomaly for PGO + 0.1Ba presented in Figure 4a unequivocally demonstrates such a shape. As the peaks in the second-order phase transition (in contrast to the first-order peaks) are usually largely extended, in most situations, it is hard to evaluate the latent heat associated with the phase transition in a qualitative way. Moreover, it was reported that an increasing amount of Ba ions [8,17] significantly affects the diffusion zone of the phase transitions in the PGO single crystals. Thus, for the samples with a higher content of Ba ions, this diffusion in combination with the second-order phase transition causes the anomaly associated with latent heat to become indistinguishable or absent.

By analyzing the absolute value of the *c*_p_ high-temperature range, it was found that the heat capacity reaches values close to 450 J/(mol·K) and agrees with the values presented in the literature [7] for PGO single crystals, pure and doped, as well as those that come from Dulong–Petit theory. The small deviations from this value may be related to different concentrations of point defects, which are usually unavoidable in technological processes. The solid lines represent the fitting of experimental data to the Einstein model [15]. Due to a significant degree of diffusion of the phase transformation and after subtracting the baseline resulting from the fit to the Einstein function, the amount of latent heat has been estimated only in two cases and is presented in Figure 5a,b. By integrating the resulting excess specific heat Δ*c*_p_(*T*) and using Equation (3), it is also possible to estimate the entropy change associated with those phase transitions, as shown in Figure 5c,d.
(3)$$∆S=\int \frac{∆c_{p}\left(T\right)}{T}dT,$$

### 3.3. Polarization Measurements

Figure 6 presents *P*(*E*) hysteresis loops measured at room temperature for PGO + xBa (x = 0.1, 0.2, 0.25, and 0.3), whereas Figure 7 shows a set of *P*(*E*) hysteresis loops for different temperatures for all of the crystals under consideration. For comparison purposes, in Figure 8 we present pure PGO *P*(*E*) hysteresis loops (also for different temperatures), which were presented in the study of PGO doped by Cr ions [10].

As was mentioned in the Materials and Methods section, the *P*(*E*) hysteresis loops have been measured at several temperatures ranging from room temperature (RT) up to a temperature higher than *T*_C_, which means also in a paraelectric phase (we have indicated that the *T*_C_ of PGO + 0.1Ba, PGO + 0.2Ba, PGO + 0.25Ba, and PGO + 0.3Ba are 425 K, 388 K, 370 K, and 348 K, respectively, so the range of the measurement temperature was appropriately higher).

The saturated *P*(*E*) loops clearly show the crystals’ pure ferroelectric character, indicating that all the presented loops exhibit distinct ferroelectric behavior. Simply speaking, the shape of the loops at room temperature is rather square. The square shape of the loops indicates that all domains are completely oriented by the electric field. With increasing temperature, *P*(*E*) loops transform into slanted loops. Slanted loops are an indication that not all domains are oriented by the electric field. Thermal vibrations of the lattice, which are responsible for that, cause fluctuations in polarization. With further increases in temperature, *P*(*E*) loops transform their shape into slim ones, which is typical behavior for that kind of material at high temperatures [1,2,10].

Regarding pure PGO at room temperature, the values of coercive field *E*_c_, remnant polarization *P*_r_, and spontaneous polarization *P*_s_ are 2.5 × 10^5^ V/m, 0.032 C/m^2^ (*E* = 0 V/m), and 0.044 C/m^2^ (*E* = 5.3 × 10^5^ V/m), respectively. The current values, along with those presented in [10], are consistent with the literature data [22]. To make it easy to compare the values of *E*_c_, *P*_r_, and *P*_s_ at room temperature for all four Ba-doped crystals, we have collected and presented the data in Table 2. An analysis of the following values of *E*_c_, *P*_r_, and *P*_s_ gives information on the transparent effect of the doping degree with Ba ions. The addition of Ba ions significantly affects not so much the value of polarization—which gradually decreases—but the value of the coercive field. The clear effect of doping also shows a decrease in electrical conductivity at high temperatures compared with pure PGO. The exception, unfortunately, is the PGO + 0.2Ba crystal, for which an increased conductivity was noted. This is related to the different colors of the crystal, i.e., an increased number of defects, as already mentioned in Section 2. The decreasing value of polarization is a consequence of nonstoichiometric effects, as described in [8]. The authors of [8] have established that disorder effects caused by Ba doping showed non-monotonic changes in crystal lattice parameters. It has been reported that PGO + 0.1Ba exhibited the highest lattice parameters, i.e., the largest unit cell. Those parameters influence the electric properties of Ba-doped PGO crystals.

### 3.4. Electrocaloric Properties

Figure 7 shows the *P*(*E*) hysteresis loops as a function of temperature, which was used to evaluate the magnitude of the ECE using the indirect method. The results of the conversion of the *P*(*E*) hysteresis loops into the *P*(*T*) curves (i.e., the temperature dependence of polarization) for all the compositions are presented in Figure 9. We would like to emphasize here that for visual purposes, it is more convenient to use the kV/cm units instead of V/m. Therefore, starting from Figure 9, the values of the electric field are expressed in kV/cm at a conversion rate of 1 × 10^5^ V/m being equal to 1 kV/cm.

Polarization gradually decreases with increasing temperature, as shown in Figure 9. It can be explained by the crystal lattice thermal vibrations, which progressively break up the ordering of dipoles as they approach the phase transition. Furthermore, a higher Ba ion content results in a wider temperature range where polarization is decreasing (i.e., the slope of the polarization lowering is highest for a PGO + 0.1Ba than for other concentrations). As seen in the *ε*(T) dependences, this indicates the diffusion of the phase transition as the amount of Ba ions increases.

Figure 10 shows the temperature dependence of the entropy change for four different compositions at an applied electric field of *E* = 12 kV/cm, whereas Figure 11 presents the evolution of the ECE for the same crystals as a function of temperature measured for different electric fields in the range of 0.5–12 kV/cm by the employed method.

The revealed peaks (maxima) in both the entropy change Δ*S* and the adiabatic change in electrocaloric temperature Δ*T*, significantly intensified by the electric field, are caused by the phase transition (it can be confronted with the DSC and permittivity results presented in Figure 2a and Figure 4 that they are a few degrees above the ferroelectric-paraelectric phase transition).

Due to the fact that the Δ*S* and Δ*T* are related by the *T*·Δ*S* = *C*·Δ*T* (where *T* and *C* are respectively temperature and heat capacity), the distribution of variable values is consistent.

If we compare the above data with those obtained from calorimetric measurements (Figure 5c,d), one can see that only the sample PGO + 0.1Ba shows comparable values. The deviation for the other crystals can be explained by the action resulting from the application of a strong electric field as well as the strong uncertainty in the selection of the integral range resulting from the significant diffusing of the phase transformation in the more heavily doped PGO crystals.

We have collected in Table 3 the values of Δ*T* obtained for pure PGO [10] and the Ba-doped crystals as well as the Cr-doped crystal [10]. Compared with the value Δ*T* = 1.2 K obtained for pure PGO at 5.3 kV/cm [10], the values of Δ*T* obtained for the Ba-doped crystals are lower. For PGO + 0.1Ba, PGO + 0.2Ba, PGO + 0.25Ba, and PGO + 0.3Ba, and for the 12 kV/cm, Δ*T* is equal to 0.18 K, 0.09 K, 0.12 K, and 0.09 K, respectively. As shown in Figure 9, increasing Ba ion concentration decreases polarization, so this Δ*T* characteristic (i.e., lowering the value of Δ*T*) was predictable. However, a very interesting feature of the influence of the Ba ions can also be seen when considering the width of the peak. For the higher-doped samples (PGO + 0.3Ba), there is a larger temperature window that could potentially be useful for micro-refrigeration applications.

In addition to the ECE calculation, we present in Figure 12 the electrocaloric temperature change as a function of the electric field at the maximum Δ*T* for all the compositions. This approach allows the comparison of the Δ*T* enhancement concerning the value of the applied electric field for different compositions. The largest increase of Δ*T* is observed for PGO + 0.1Ba, whereas the increase of ΔT for other crystals is very similar and almost parallel. There also exists a more visible deviation for PGO + 0.2Ba, which has already been noted in other measurements.

As reported in [8,9], Ba ions induce the appearance of local polar nanoregions (LPNR), which is the reason for this significant diffusing of the phase transition. LPNR manifests in incomplete compensation of hysteresis loops above the *T*_C_ temperature, leading to nonzero macroscopic polarization in the wide temperature region above *T*_C_. A characteristic “tail” in a paraelectric phase is observed for the temperature dependence of the spontaneous polarization obtained from the hysteresis measurements. This incomplete compensation of local polarization is usually provoked by microscopic inhomogeneities present in the doped PGO single crystal. Such inhomogeneities in the form of the local disorder with anisotropy effects are related to Ba ion doping, oxygen vacancies, and Ge-ion precipitation, as shown in [8] based on XPS research.

Increasing Ba doping considerably influences the values of polarization and the coercive field. Decreasing polarization influences the decrease of the overall electrocaloric effect, although the diffusing of the phase transition increases the range of the effect’s occurrence, which might be beneficial in applications.

Electrocaloric properties are more advantageous for the small degree of doping with Ba ions in PGO. In all studies, a clear deviation for the PGO + 0.2Ba crystal has been observed. This is most likely due to the larger nonstoichiometry of this particular composition (darker coloration of the single crystals when compared with other compositions), as already mentioned in this work in Section 3.1.

## 4. Conclusions

We have made systematic measurements of some ferroelectric properties of lead germanate single crystals doped with Ba ions. The main conclusion that can be drawn from the measurements presented is that the addition of Ba has a significant influence on both the basic properties and the phase transitions in PGO + xBa single crystals. We are still dealing with a second-order phase transition; however, the characteristic gradual diffusing (with increasing Ba content) of the phase transition also occurs in PGO + xBa single crystals. The Curie –Weiss law starts to be valid as early as several degrees above the temperature of the phase transition (see Figure 2b). The influence of Ba doping on the ECE is rather unfavorable. However, the fact that the phase transformation is diffused expands the temperature range of occurrence of this phenomenon.

## Figures and Tables

**Figure 1 materials-16-00413-f001:**
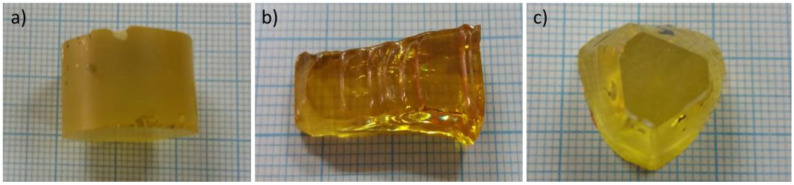
Photographs of the as-grown crystals (**a**) PGO + 0.1Ba, (**b**) PGO + 0.2Ba, (**c**) PGO + 0.25Ba.

**Figure 2 materials-16-00413-f002:**
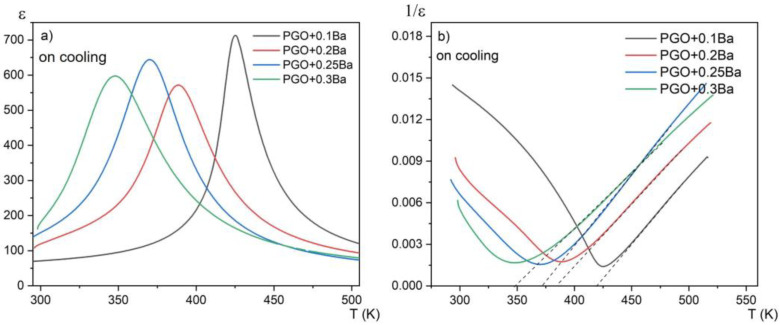
(**a**) temperature dependences of the real part of the linear dielectric susceptibility for PGO + 0.1Ba, 0.2Ba, 0.25Ba, and 0.3Ba; (**b**) temperature dependences of the reciprocal of dielectric susceptibility for PGO + 0.1Ba, 0.2Ba, 0.25Ba, and 0.3Ba on cooling. Dashed lines represent compliance with the Curie–Weiss law.

**Figure 3 materials-16-00413-f003:**
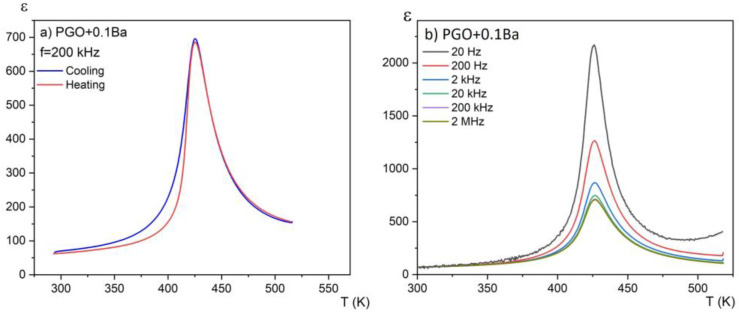
Temperature dependences of the real part of the linear dielectric susceptibility for PGO + 0.1Ba (**a**) on the heating and cooling modes; (**b**) for a set of different frequencies on cooling.

**Figure 4 materials-16-00413-f004:**
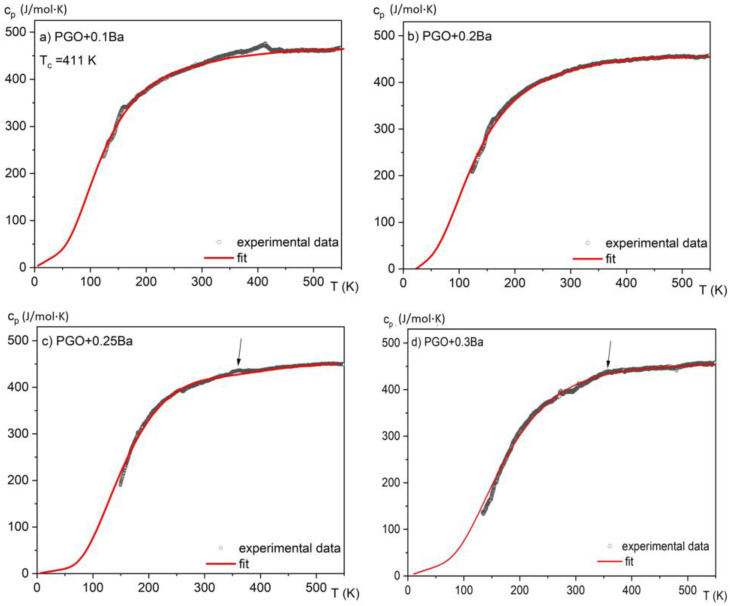
Temperature dependences of the specific heat for different compositions of PGO + xBa crystals: (**a**) PGO + 0.1Ba, (**b**) PGO + 0.2Ba, (**c**) PGO + 0.25Ba, (**d**) PGO + 0.3Ba. The arrows point to weak but visible anomalies connected to the phase transition for PGO + 0.25Ba and PGO + 0.3Ba crystals.

**Figure 5 materials-16-00413-f005:**
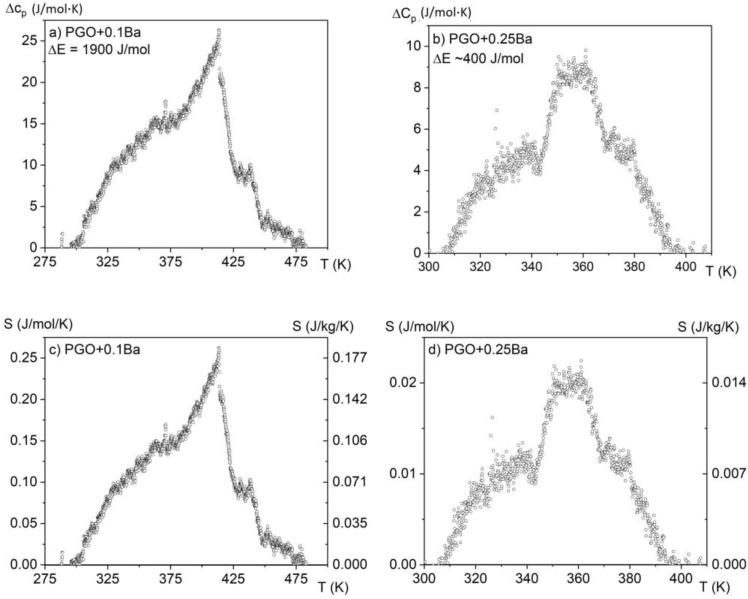
Excess specific heat and estimated values of latent heat (**a**,**b**) and entropy change (**c**,**d**) for two samples: PGO + 0.1Ba and PGO + 0.25Ba.

**Figure 6 materials-16-00413-f006:**
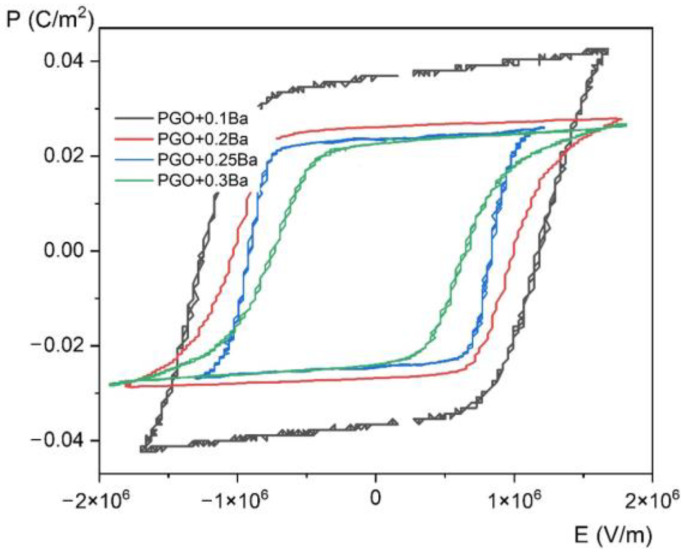
Room temperature *P*(*E*) (polarization-electric fields) hysteresis loops of PGO + xBa (x = 0.1, 0.2, 0.25, and 0.3).

**Figure 7 materials-16-00413-f007:**
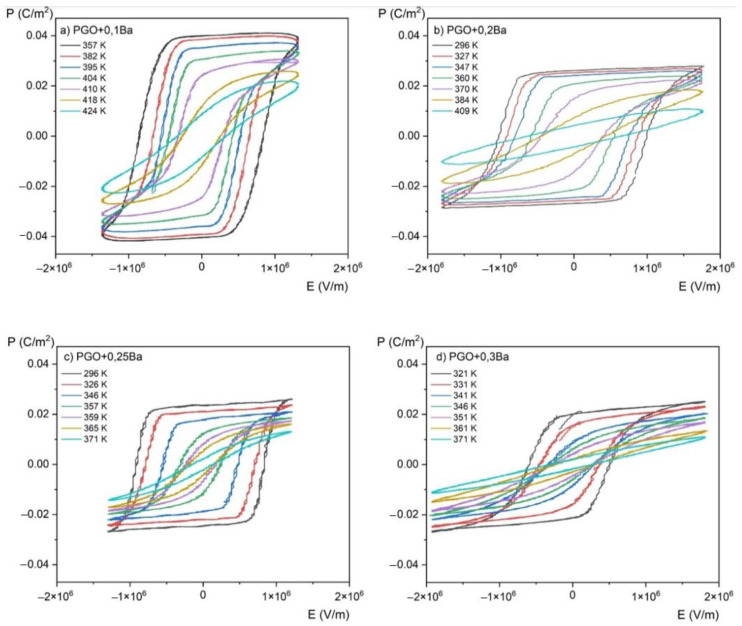
*P*(*E*) (polarization-electric fields) hysteresis loops as a function of the temperature for PGO + xBa crystals: (**a**) PGO + 0.1Ba, (**b**) PGO + 0.2Ba, (**c**) PGO + 0.25Ba, (**d**) PGO + 0.3Ba.

**Figure 8 materials-16-00413-f008:**
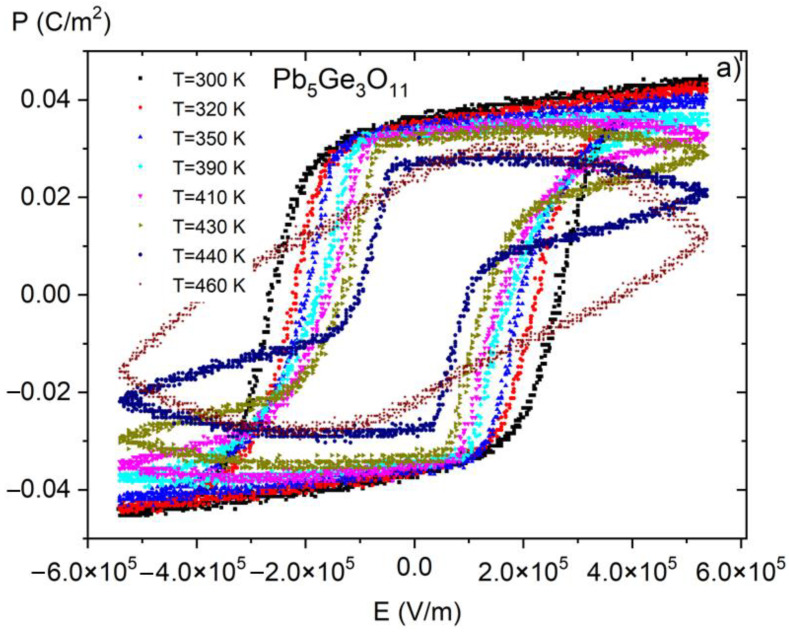
*P*(*E*) (polarization-electric fields) hysteresis loops as a function of temperature for pure PGO crystal. Reproduced from [10] with permission from the Royal Society of Chemistry.

**Figure 9 materials-16-00413-f009:**
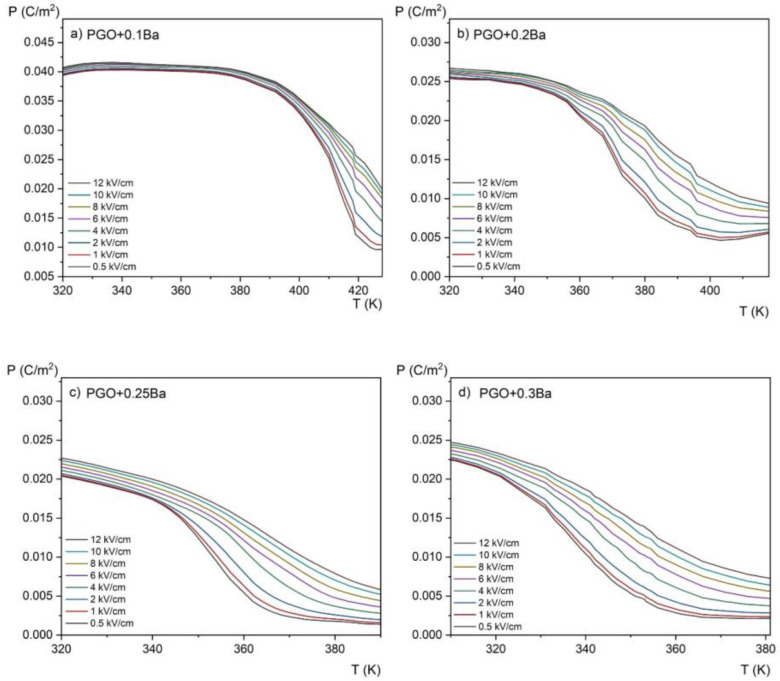
Temperature dependence of the spontaneous polarization measured at different electric fields in the range of 0.5–12 kV/cm for PGO + xBa crystals: (**a**) PGO + 0.1Ba, (**b**) PGO + 0.2Ba, (**c**) PGO + 0.25Ba, (**d**) PGO + 0.3Ba.

**Figure 10 materials-16-00413-f010:**
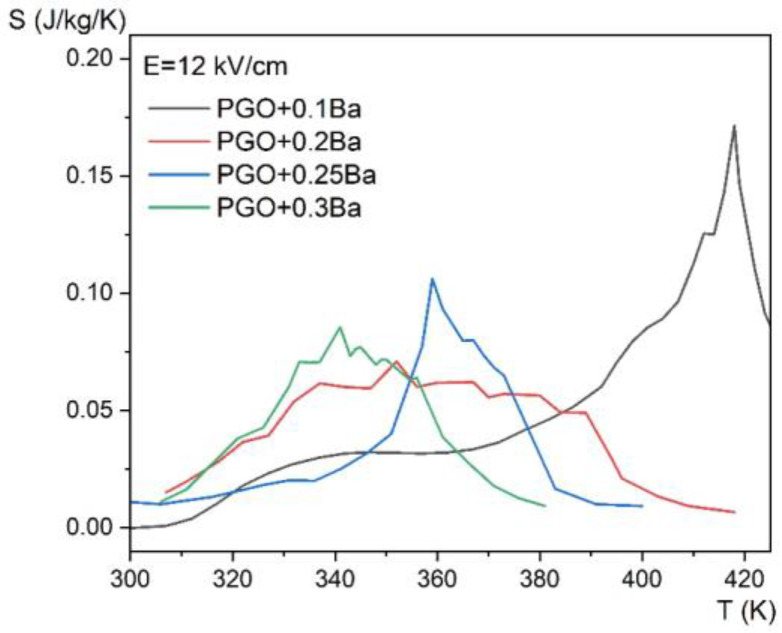
Entropy change as a function of temperature, for *E* = 12 kV/cm for PGO + xBa crystals: PGO + 0.1Ba, PGO + 0.2Ba, PGO + 0.25Ba, PGO + 0.3Ba.

**Figure 11 materials-16-00413-f011:**
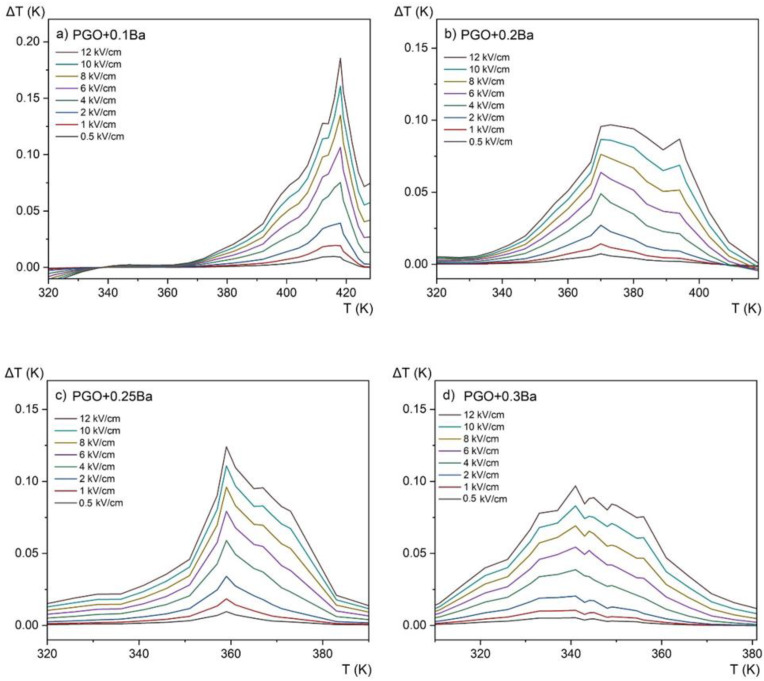
Electrocaloric temperature change (Δ*T*) as a function of temperature at 0.5–12 kV/cm applied fields for PGO + xBa crystals: (**a**) PGO + 0.1Ba, (**b**) PGO + 0.2Ba, (**c**) PGO + 0.25Ba, (**d**) PGO + 0.3Ba.

**Figure 12 materials-16-00413-f012:**
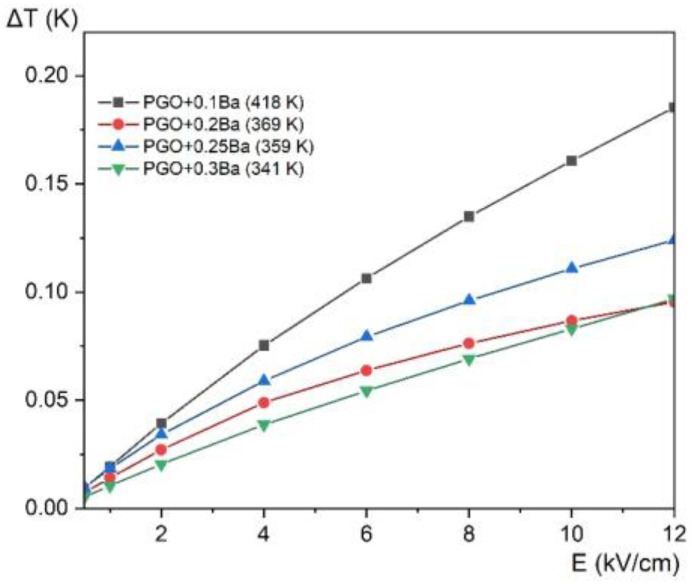
Electrocaloric temperature change (Δ*T*) as a function of the electric field at the maximum Δ*T* temperature (given in the parenthesis in the legend) for PGO + 0.1Ba, 0.2Ba, 0.25Ba, and 0.3Ba.

**Table 1 materials-16-00413-t001:** A comparison of the values of Curie temperature *T*_C_, Curie–Weiss temperature *T*_O_, and Curie–Weiss constant *C*, and maximum value of electric permittivity ε_max_ for the PGO + xBa (x = 0.1, 0.2, 0.25, and 0.3) as well as pure PGO.

Composition	*T*_C_ [K]	*T*_O_ [K]	*C* [K]	*ε_max_* (20 kHz)
PGO [10]	453	453	9.8·10^3^	3600
PGO + 0.1Ba	425	420	1.00·10^4^	700
PGO + 0.2Ba	388	385	1.11·10^4^	570
PGO + 0.25Ba	370	371	9.73·10^3^	650
PGO + 0.3Ba	348	348	1.28·10^4^	600

**Table 2 materials-16-00413-t002:** A comparison of the values of coercive field *E*_c_, remnant polarization *P*_r_, and spontaneous polarization *P*_s_ for the PGO + xBa (x = 0.1, 0.2, 0.25, and 0.3) collected at RT.

Composition	*E*_C_ [V/m]	*P*_r_ [C/m^2^] for *E* = 0 V/m	*P*_s_ [C/m^2^] for *E* [V/m]
PGO [10]	2.7 × 10^5^	0.037	0.044 (*E* = 5.3 × 10^5^ )
PGO + 0.1Ba	10.5 × 10^5^	0.036	0.042 (*E* = 12.2 × 10^5^ )
PGO + 0.2Ba	10 × 10^5^	0.026	0.027 (*E* = 17.7 × 10^5^ )
PGO + 0.25Ba	8.4 × 10^5^	0.023	0.026 (*E* = 12.1 × 10^5^ )
PGO + 0.3Ba	5 × 10^5^	0.022	0.025 (*E* = 18 × 10^5^ )

**Table 3 materials-16-00413-t003:** A comparison of the values of the maximum electrocaloric temperature change ΔT under the electric field E for the PGO + xBa (x = 0.1, 0.2, 0.25, and 0.3), PGO, and PGO + xCr (x = 0.2).

Composition	ΔT [K]	E [kV/cm]
PGO + 0.1Ba	0.18	12
PGO + 0.2Ba	0.09	12
PGO + 0.25Ba	0.12	12
PGO + 0.3Ba	0.09	12
PGO [10]	1	5.3
PGO + 0.2Cr [10]	1.5	6

## Data Availability

The data presented in this study are available on request from the corresponding author.

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
