# Peer review of "Composition-Related Dielectric, Ferroelectric and Electrocaloric Properties of Pb5Ge3O11 Single Crystals Modified by Ba Ions"

_materials, 2023, doi:10.3390/ma16010413_

Round 1

Reviewer 1 Report

In the article Electrocaloric effect and dielectric properties in uniaxial ferroelectric single crystals of Pb5Ge3O11 modified by Ba ions, the change in the properties of the Pb5Ge3O11 single crystal modified by Ba ions is considered, in particular, the change in dielectric, thermodynamic properties depending on the modification conditions is considered. In general, this direction is quite promising and interesting not only from the point of view of fundamental knowledge about the properties of Pb5Ge3O11, but also from the point of view of practical significance, which consists in assessing the applicability of the proposed modification method. The authors use a fairly large number of different research methods to characterize the observed changes, as well as interpret the obtained dependencies. This article corresponds to the subject of the declared journal and can be accepted for publication after the authors answer a number of questions that the reviewer has during its analysis.

1. The observed changes in the dielectric properties are quite interesting and informative, but the authors should give the values of the maxima of linear dielectric susceptibility depending on the concentration of barium ions.

2. The authors should give a more detailed description of the method of obtaining the selected Pb5Ge3O11 crystals, as well as the reasons for choosing barium as a dopant. In this case, these values were taken in weight or mass values. The authors should also indicate which starting components were chosen to obtain crystals.

3. Polarization hysteresis loops require additional explanations, in particular, what is the reason for such strong differences in the shape of the loops when going from a concentration of 0.1 to 0.2.

4. Technical comments include the quality of a number of submitted figures that need to be corrected when resubmitting the article.

5. In the abstract, the authors should give some specific details about the reasons for choosing Pb5Ge3O11 single crystals as objects of study, as well as their prospects for application in various directions. Also, in the abstract, the authors should indicate how the modification with barium ions occurs, since without a full analysis of the text it is not clear how the modification occurs.

6. The conclusion is large enough, the authors are invited to divide it into two parts and make it a section Discussion of the results.

7. Additional comments include the absence of any information about the structural properties of the synthesized single crystals, and how they change with increasing dopant concentration.

Author Response

We would like to thank the reviewers for the work they put into their reviews and their positive approach to the topic presented.

We have carefully made revisions and prepared point-to-point responses to comply with the referees’ comments. The changes have been marked using the “Track Changes” in the revised manuscript, and the point-to-point response to the comments is listed below. We hope that in its present form the manuscript meets the criteria for publication in Journal “Materials”.

Question 1: The observed changes in the dielectric properties are quite interesting and informative, but the authors should give the values of the maxima of linear dielectric susceptibility depending on the concentration of barium ions.

Answer 1: We have prepared a summary table and placed it in the text (as Table 1) so that the reader could see the changes caused by doping and their effect on the dielectric values

Question 2: The authors should give a more detailed description of the method of obtaining the selected Pb5Ge3O11 crystals, as well as the reasons for choosing barium as a dopant. In this case, these values were taken in weight or mass values. The authors should also indicate which starting components were chosen to obtain crystals.

Answer 2: The crystals used in the measurements were grown in the Royal and Signal Radar Establishment laboratory in Malvern UK. We added the literature we have included in the manuscript a literature reference to the growing details. In our study, we used the same crystals that were studied by the authors of the publication [8]. This publication contains a compositional analysis and structural studies. In the introduction part, we also included information on why we decided to study these crystals in particular (i.e. Ba dopant).

Question 3: Polarization hysteresis loops require additional explanations, in particular, what is the reason for such strong differences in the shape of the loops when going from a concentration of 0.1 to 0.2.

Answer 3: The authors of [8] have established that disorder effects caused by Ba doping showed non-monotonic changes in crystal lattice parameters. Those parameters influence the electric properties of Ba-doped PGO crystals. In [8] it was reported that PGO+0.1Ba exhibited the highest lattice parameters i.e. has the largest unit cell. Changes in electrical parameters resulting from doping and the degree of non-stoichiometry are typical for this kind of material.

Question 4: Technical comments include the quality of a number of submitted figures that need to be corrected when resubmitting the article.

Answer 4: For the purpose of review, the editor allows the placement of drawings in the text, which may not be of the highest quality, If the article is accepted, it is our responsibility to provide the editor with high-quality figures for reproduction.

Question 5: In the abstract, the authors should give some specific details about the reasons for choosing Pb5Ge3O11 single crystals as objects of study, as well as their prospects for application in various directions. Also, in the abstract, the authors should indicate how the modification with barium ions occurs, since without a full analysis of the text it is not clear how the modification occurs.

Answer 5: We have prepared the abstract according to the editor’s requirements. It should not contain more than 200 words. Present only the main objectives and should not exaggerate the main conclusions.

For this reason, we included in the abstract briefly what was studied and how  Ba doping influences all the measured parameters. We highlighted the novelty - that the ECE effect has never been measured in this material. Since our research belongs to the so-called basic research, we do not indicate the specific application of our results only the possibilities. We are investigating whether this material can have useful values of the ECE effect and other ferroelectric quantities.

Question 6: The conclusion is large enough, the authors are invited to divide it into two parts and make it a section Discussion of the results.

Answer 6: We corrected the conclusion part as suggested by the reviewer.

Question 7: Additional comments include the absence of any information about the structural properties of the synthesized single crystals, and how they change with increasing dopant concentration.

Answer 7: The general structure of the PGO single crystals was described in the introduction. Structural analysis for Pb5-xBaxGe3O11 single crystals was performed and described in the paper [8]. The addition of Ba does not change the basic structure of PGO, but it does change the size of the unit cell (see comment to point 3). A comment on this issue is included in the revised text.

Reviewer 2 Report

Attached

Author Response

We would like to thank the reviewers for the work they put into their reviews and their positive approach to the topic presented.

We have carefully made revisions and prepared point-to-point responses to comply with the referees’ comments. The changes have been marked using the “Track Changes” in the revised manuscript, and the point-to-point response to the comments is listed below. We hope that in its present form the manuscript meets the criteria for publication in Journal “Materials”.

1: The indirect method/ Maxwell’s method is not “so-called” indirect method. It is an indirect method as in this method, the ECE temperature change is not measured directly but estimated from polarization change. So, there should not be any doubt about it.

Answer 1: Thank you for this remark – we have removed the term “so-called”

  1. The abstract must be improved.

(a) For example, the description of the calculation process in the abstract is unnecessary. This can be stated in the section, Materials and Methods.

Answer 2a: In accordance with the reviewer's suggestion, we removed this description from the abstract

(b) Some of the highlightable outcomes should be quantitatively stated in the abstract.

Answer 2b: Due to the excessive number of measured parameters, we intentionally did not want to use quantitative description in the abstract. In the abstract, we have outlined what is changing and in what direction. We carry out a quantitative analysis in the improved version, in particular by presenting tables.

  1. Introduction should be improved.

(a) The title of the manuscript is “Electrocaloric effect and dielectric properties in uniaxial ferroelectric single crystals of Pb5Ge3O11 modifiedby Ba ions”. However, Electrocaloric effect, dielectric properties and proper reasons for Ba-substitution is missing in the introduction.

Answer 3a: We have improved the Introduction part to include the missing motivation for this work.

(b) Secondly, unnecessary elaboration of unrelated properties of PGO should be removed or stated in one single sentence.

Answer 4b: PGO and its related solid solution are important ferroelectric crystals that are usually used in nonlinear optics. We wanted to present some basic properties of this crystal and those for which it is used in the applications. On the other hand, there are some properties that have never been measured for these crystals. That is why we made maybe a little extended introduction. However to satisfy the reviewer we have slightly shortened the first paragraph of the introduction.

(c) the recent developments in ECE should be mentioned such as the reliability of ECE, etc.

Answer 3c: We have added a paragraph on the motivation for our measurements of the ECE effect.

  1. [Fig 1] PGO+0.3Ba is missing.

Answer 4: Yes indeed. As we mentioned that our crystals are the same as that used in previous measurements by the authors of [8 and 9]. In the case of PGO+0.3Ba, we had a small piece of a crystal whose color was the same as PGO+0.1Ba and PGO+0.25 Ba. Unfortunately, it has already been cut into smaller slices and we can no longer call it as-grown.

  1. [Line 109-113] “The addition of barium ions into PGO causes a gradual (as Ba content increases) diffusion of the phase transformation as well as decreasing value of ε at the TC point (the exception is PGO+0.2Ba which shows inferior results probably related to the greater defectiveness of the crystal - darker coloration – due to increased number of so called color centres).” Do the authors have any experimental evidence to show in this regard?

Answer 5: We have explained this problem on the basis of earlier XPS studies contained in the paper [8] and the analysis contained in the paper [16]

  1. [Fig 6, Line 188-201, Table-1] (a) PGO+0.1Ba shows a drastic change from the PGO reference data. What is the reason? (b) If the addition of 0.1Ba to PGO enhances Pr and EC, then, why do these decrease with further additions (i.e. for PGO+0.2Ba, PGO+0.25Ba, PGO+0.3Ba)?

Answer 6: Let me repeat the answer which has already been included i.e. point 3 for Reviewer 1.

The authors of [8] have established that disorder effects caused by Ba doping showed non-monotonic changes in crystal lattice parameters. Those parameters influence the electric properties of Ba-doped PGO crystals. In [8] it was reported that PGO+0.1Ba exhibited the highest lattice parameters i.e. has the largest unit cell. Changes in electrical parameters resulting from doping and the degree of non-stoichiometry are typical of these kinds of materials.

  1. The ECE temperature change is very small compared to other materials. ECE strength of these materials should be calculated and compared with other reported materials in a table.

Answer 7: We have compared our current results which we measured earlier for PGO doped with Cr ions. The data are collected i.e. table 3.

  1. In this series of PGO+xBa, undoped PGO could be added so that they can be compared on the same experimental condition and the influence of the Ba can be understood well.

Answer 8: These data are collected in table 3

  1. Recent related works are missing in the discussion and introduction.

Answer 9:We have updated some literature points

  1. The conclusion appears to be a part of the discussion. It explains a few facts but does not conclude the work. It must be improved.

Answer 10: We improved the conclusions according to the Reviewer’s suggestions. We moved the part of conclusions to section 3. Results and discussion.

Reviewer 3 Report

In the present work, authors reported the synthesis of single crystals of Pb5Ge3O11 modified by Ba ions, and then investigated their dielectric and ferroelectric properties. Results indicated that increasing Ba doping influenced all the measured parameters, especially by diffusing the phase transition and decreasing values of polarization as well as the coercive field. A series of results were discussed. However, the innovation in this paper is not very well put forward, some issues should be addressed.

1, The abstract can be polished and improved. The novelty problem statement described by the authors should be emphasized to attract general readers by providing more insights on the experimental observations. Also, the authors should elaborate the general applicability of the current work.

2, The introduction writing part need to be improved. Also, the writing and presentation of the introduction lacks a bit in clarity. The paper requires some amount of rewriting to clarify all aspects of it, especially the novelty and new findings of this work that need to be clearly mentioned. The authors have mentioned " However, there are few studies on the electro-thermodynamic properties of these materials...."…if this is the motivation of the current work, this point needs to be elaborated with existing research work aligned to this direction.

3, Authors may rearrange/polish the text and elaborated " Materials and Methods" section the way so anybody can repeat the procedures, like a recipe. If there is process flow diagram can be added in fig. 1, it would be helpful to non-specialist readers (optional).

4, What is percentage of Ba in the Pb5-xBaxGe3O11 particles? How to confirm the x value in Pb5-xBaxGe3O11. The control of contents of doping Ba is very important. I am wondering how its content is controlled and how the content of doping Ba. I suggest you make a component analysis, such as spectroscopic analysis or X-ray dispersed spectroscopic. The results are crucial for confirming the dielectric and other properties.

5, Some key and important research results in dielectric field should be mentioned and cited so that we can provide a solid background and progress to the readers, such as Materials, 2022, 15, 8816;Journal of Materials Chemistry C, 2016, 4, 9738; ACS Applied Materials & Interfaces, 2017, 9, 16404.

6, It was unable to provide a satisfactory modelling of the dielectric mechanism (s): To what is due dielectric (polarization, relaxation, resonance, interfacial polarization, defect polarization, dipole ...)? This fundamental issue is not all answered. Please revise this section and clarify the mechanism.

7, In addition, it was said that “Hence this dispersion can presumably be linked to the existence of defects. Intriguingly for a higher doped lead germinate single crystals the strength of this dispersion considerably decreases.” It is better to refer to the existing literatures to support the description. Does this result mean that defect-induced dipole polarization should be responsible for the dielectric property variation.

8, In Table, the variation trend of Ba doped PGO showed an odd tendency in Pr values. Please check the Pr values.

9, Please refine the references, and update the up-to-date references.

Author Response

We would like to thank the reviewers for the work they put into their reviews and their positive approach to the topic presented.

We have carefully made revisions and prepared point-to-point responses to comply with the referees’ comments. The changes have been marked using the “Track Changes” in the revised manuscript, and the point-to-point response to the comments is listed below. We hope that in its present form the manuscript meets the criteria for publication in Journal “Materials”.

1. The abstract can be polished and improved. The novelty problem statement described by the authors should be emphasized to attract general readers by providing more insights on the experimental observations. Also, the authors should elaborate the general applicability of the current work.

Answer 1: We have corrected and improved the abstract. The novelty is that the ECE effect has never been measured in this material. However, since our research belongs to the so-called basic research, we do not indicate the specific application of our results only the possibilities. We are investigating whether this material can have useful values of the ECE effect and other ferroelectric quantities.

2. The introduction writing part need to be improved. Also, the writing and presentation of the introduction lacks a bit in clarity. The paper requires some amount of rewriting to clarify all aspects of it, especially the novelty and new findings of this work that need to be clearly mentioned. The authors have mentioned "However, there are few studies on the electro-thermodynamic properties of these materials...."…if this is the motivation of the current work, this point needs to be elaborated with existing research work aligned to this direction.

Answer 2: Thank you for this remark. We have improved the Introduction part to better describe the objectives of our research

3. Authors may rearrange/polish the text and elaborated " Materials and Methods" section the way so anybody can repeat the procedures, like a recipe. If there is process flow diagram can be added in fig. 1, it would be helpful to non-specialist readers (optional).

Answer 3: We have described research methods with better precision so that it is possible to repeat our measurements. The present description should be well understood by the ferroelectric materials research community

4. What is percentage of Ba in the Pb5-xBaxGe3O11 particles? How to confirm the x value in Pb5-xBaxGe3O11. The control of contents of doping Ba is very important. I am wondering how its content is controlled and how the content of doping Ba. I suggest you make a component analysis, such as spectroscopic analysis or X-ray dispersed spectroscopic. The results are crucial for confirming the die.

Answer 4: The exact chemical composition and possible deviations from stoichiometry for these crystals have already been measured using the XPS method and structural studies and described in the paper [8]. We have used for our research already structurally developed crystals. It was not necessary to repeat these measurements. The chemical composition i.e. x value is correct.

5. Some key and important research results in dielectric field should be mentioned and cited so that we can provide a solid background and progress to the readers, such as Materials, 2022, 15, 8816;Journal of Materials Chemistry C, 2016, 4, 9738; ACS Applied Materials & Interfaces, 2017, 9, 16404.

Answer 5: Thank you for this suggestion however we note with regret that the subject matter of the presented papers to be quoted does not very much match the results presented in our manuscript.

6. It was unable to provide a satisfactory modelling of the dielectric mechanism (s): To what is due dielectric (polarization, relaxation, resonance, interfacial polarization, defect polarization, dipole ...)? This fundamental issue is not all answered. Please revise this section and clarify the mechanism.

Answer 6: We do not deal with the mechanism of dielectric dispersion in this work. We only note that it exists and that it does not have a relaxation character, so there is no reason to study and model it further. Our goal was to measure the effect of doping on the ferroelectric properties of PGO single crystals with special attention to the electrocaloric effect.

7. In addition, it was said that “Hence this dispersion can presumably be linked to the existence of defects. Intriguingly for a higher doped lead germinate single crystals the strength of this dispersion considerably decreases.” It is better to refer to the existing literatures to support the description. Does this result mean that defect-induced dipole polarization should be responsible for the dielectric property variation.

Answer 7: As mentioned earlier dielectric dispersion is not the subject of this paper. Nevertheless, as already mentioned in this manuscript, a certain oxygen deficit is always present in PGO crystals. It results from technological conditions. doping with Ba ions can compensate for this deficit, hence the observed decrease in dispersion. This topic may be interesting for future considerations

8. In Table, the variation trend of Ba doped PGO showed an odd tendency in Pr values. Please check the Pr values.

Answer 8: We have checked the values of Pr and Ps and make some corrections in the table.

 9. Please refine the references, and update the up-to-date references.

Answer 9: We added some references according to the suggestions of the Reviewer.

Round 2

Reviewer 1 Report

The authors answered all the questions, the article can be accepted for publication.

Author Response

We would like to thank the reviewers for their thoughtful comments and efforts towards improving our manuscript. We appreciate that the answers have been accepted. To improve the quality of our manuscript, it has been corrected one more time by a native speaker.

Reviewer 2 Report

The authors have improved the manuscript and now it can be accepted.

Author Response

(The authors gave the same response as above.)

Reviewer 3 Report

There are still unsovled issues 1. the compositions of particles were still missing and XPS cannot confirm the composition of matters. 2. References were still outdated.

Author Response

(The authors gave the same response as above.)

Round 3

Reviewer 3 Report

All issues were well addressed, and this work can be accepted. However, It is not suggested to applied XPS to charater the composition of crystals since the XPS is a tool for analyzing the surfacial state of matters and cannot reflect the real content of atoms.

Author Response

(The authors gave the same response as above.)
